Manuscript prepared for Geosci. Model Dev.
with version 2015/09/17 7.94 Copernicus papers of the LaTeX class copernicus.cls.
Date: 26 October 2016

# PhytoSFDM version 1.0.0: Phytoplankton Size and Functional Diversity Model

Esteban Acevedo-Trejos[1], Gunnar Brandt[1,2], S. Lan Smith[3], and
Agostino Merico[1,4]

[1]Systems Ecology Group, Leibniz Center for Tropical Marine Ecology, Fahrenheitstrasse 6, 28359
Bremen, Germany
[2]Current address: Brockmann Consult, Max-Planck-Str. 7, 21052 Geesthacht, Germany
[3]Marine Ecosystem Dynamics Research Group, Research and Development Center for Global
Change, Japan Agency for Marine-Earth Science and Technology, Yokohama, Japan
[4]Faculty of Physics & Earth Sciences, Jacobs University Bremen, Campus Ring 1, 28759 Bremen,
Germany

*Correspondence to:* Esteban Acevedo-Trejos (esteban.acevedo@leibniz-zmt.de)

**Abstract.** Biodiversity is one of the key mechanisms that facilitate the adaptive response of plank-
tonic communities to a fluctuating environment. How to allow for such a flexible response in ma-
rine ecosystem models is, however, not entirely clear. One particular way is to resolve the natural
complexity of phytoplankton communities by explicitly incorporating a large number of species
or plankton functional types. Alternatively, models of aggregate community properties focus on
macroecological quantities such as total biomass, mean trait, and trait variance (or functional trait
diversity), thus reducing the observed natural complexity to a few mathematical expressions. We
developed the modelling tool PhytoSFDM, which can resolve species discretely and can capture ag-
gregate community properties. The tool also provides a set of methods for treating diversity under
realistic oceanographic settings. This model is coded in Python and is distributed as an open-source
software. PhytoSFDM is implemented in a 0D physical scheme and can be applied to any location
of the global ocean. We show that aggregate-community models reduce computational complex-
ity while preserving relevant macroecological features of phytoplankton communities. Compared to
species-explicit models, aggregate models are more manageable in terms of number of equations
and have faster computational times. Further developments of this tool should address the caveats
associated with the assumptions of aggregate community models and on implementations into spa-
tially resolved physical settings (1D and 3D). With PhytoSFDM we embrace the idea of promoting
open source software and encourage scientists to build on this modelling tool to further improve our
understanding of the role that biodiversity plays in shaping marine ecosystems.

## 1  Introduction

Numerical models are simplified abstractions of complex phenomena. They are engineered for the problem at hand and cannot be designed to maximize simultaneously the three key requirements of generality, precision, and realism, because one of these must be sacrificed in favour of the other two (Levins, 1966). Marine ecosystem models are no exceptions, and the scientific community has questioned the trend towards increasing model complexity, in terms of large numbers of state variables and parameters (Fulton et al., 2003; Anderson, 2005; Hood et al., 2006; Anderson, 2010). Alternatives such as trait-based models have been put forward as a way to simplify overly parameterised ecosystem models (Follows and Dutkiewicz, 2011).

In the past two decades, trait-based models of planktonic ecosystems have become important tools for elucidating the fundamental mechanisms behind emergent patterns of community structure and diversity. Most of these models describe the phytoplankton community by a *discrete* representation of many species or functional groups (Baird and Suthers, 2007; Follows et al., 2007; Bruggeman, 2007; Barton et al., 2010; Banas, 2011; Ward, 2012; Smith et al., 2015). Alternatively, models have been developed that treat the whole phytoplankton species assemblage as a single entity (Wirtz and Eckhardt, 1996; Norberg et al., 2001; Merico et al., 2009; Bruggeman, 2009; Wirtz, 2013; Wirtz and Sommer, 2013; Terseleer et al., 2014; Acevedo-Trejos et al., 2015). These models use *aggregate* community properties such as total biomass, mean trait, and trait variance to describe changes in phytoplankton community composition. Hence, by approximating the full spectrum of species or functional types with just a few macroecological properties, these models present a way of reducing the complexity of natural communities (Merico et al., 2009).

The simplification of both types of trait-based models (i.e. discrete and aggregate) relies on the use of a key trait, for which relationships with other traits can be formulated. Cell size is recognised as one of the most important traits for characterising phytoplankton communities (Litchman et al., 2008; Finkel et al., 2010; Litchman et al., 2010; Marañón, 2015), and it has been commonly used in plankton ecosystem models (Baird and Suthers, 2007; Banas, 2011; Ward, 2012; Wirtz, 2013; Wirtz and Sommer, 2013; Terseleer et al., 2014; Acevedo-Trejos et al., 2015; Smith et al., 2015). This morphological trait affects trophic organisation of foodwebs and the sequestration of $CO_2$ into the ocean interior (Chisholm, 1992). Phytoplankton size also impacts on many ecological and physiological functions and is linked to other relevant traits via tradeoff relationships (see reviews by Litchman et al., 2008; Finkel et al., 2010; Litchman et al., 2010). Therefore, studies on how cell size is associated to ecological and physiological processes and on the impact that these associations have on the structure and functioning of planktonic communities are of fundamental importance (Marañón, 2015; Andersen et al., 2015).

Here we present a new phytoplankton size and functional diversity model (called PhytoSFDM) that allows for five different ways of describing the size composition of phytoplankton communities in the upper mixed layers of the world oceans. In the first variant, the phytoplankton community is

described according to the classical approach that resolves the discrete assemblage of many different species and then we present four alternative ways to express aggregate community properties of phytoplankton based on four different ways of treating size diversity. We provide this model as open-source so that it can be used, modified and redistributed freely with the aims of fostering reproducibility and encouraging investigations about the impact of environmental conditions on properties of phytoplankton community structure and diversity.

## 2 Model description

PhytoSFDM is developed from the study of Acevedo-Trejos et al. (2015), which used a size-based model of aggregate community properties to investigate the phytoplankton size structure and size diversity in two environmentally contrasting regions of the Atlantic Ocean. In this model, the phytoplankton community self-assembles according to a trade-off emerging from relationships between cell size and (1) nitrogen uptake, (2) zooplankton grazing, and (3) phytoplankton sinking. In PhytoSFDM we have extended this work by providing four ways of treating size diversity using a moment-based approximation (see Smith et al., 2011; Bonachela et al., 2015, and section 2.1.3 in this study). In addition, we include a discrete version of the model (hereafter referred to as the full model) to better illustrate the potential of using aggregate models as compared to the equivalent discrete version. In the following, we present the mathematical equations, a description of the code structure, and easy-to-follow examples of how to use the model.

### 2.1 Mathematical formulations

#### 2.1.1 Mixed layer scheme

The zero-dimensional physical setup consists of two vertical layers, the upper-mixed layer containing the pelagic ecosystem and the abiotic bottom layer with nitrogen concentration as forcing. Following Evans and Parslow (1985) and Fasham et al. (1990), we describe material exchange between the two layers (K) as a function of the mixed-layer depth (M),

$$K = \frac{\kappa + h^+(t)}{M(t)}, \tag{1}$$

where $\kappa$ is a constant that parameterises diffusive mixing across the thermocline and $h^+(t)$ is a function that describes entrainment and detrainment of material. The latter is given by $h^+(t) = max[h(t), 0]$, with $h(t) = dM(t)/dt$.

Zooplankton are considered capable of maintaining themselves within the upper mixed layer, thus, their mixing term simplifies to $K_Z = h(t)$.

### 2.1.2 Dynamics of the full phytoplankton community

The description of the phytoplankton community is a trait-based variant of the classical Nutrient-Phytoplankton-Zooplankton-Detritus (NPZD) model (Fasham et al., 1990). We consider only one nutrient, nitrogen, which constitutes the currency of our model, one zooplankton population (composed of individuals assumed to be identical) and a single detritus pool. We define n morphologically distinct phytoplankton types (hereafter referred to as morphotypes), and we consider n equal to either 10 or 100. Each morphotype is characterised by a biomass $P_i$ and a cell size $S_i$, in units of $\mu$m Equivalent Spherical Diameter. The distribution of biomass along the size dimension is known to be positively skewed (i.e. an asymmetrical size distribution with a pronounced right tail compared to its left tail), due to physiological, morphological and ecological constraints that limit phytoplankton from a minimum size of around 0.15 $\mu$m ESD to a maximum size of about 575 $\mu$m ESD (Marañón, 2015; Andersen et al., 2015). Consequently, we assume a log-normal distribution of size to represent the size of each morphotype, thus transforming the cell size $S_i$ as follows $L_i = \ln(S_i)$. The net growth rate of the whole phytoplankton community (P) is then given by:

$$\frac{dP}{dt} = \sum_{i=1}^{n} f_i(L_i, E) \cdot P_i,$$ (2)

where $f_i(L_i, E)$ is the net growth rate of size class i, which we assume to be a proxy for fitness (Smith et al., 2011). Hence $f_i$ accounts for the gains and losses of each morphotype as a function of cell size ($L_i$) and environment E. The latter includes changes in nitrogen, irradiance, temperature, and grazing. The equation describing the fitness functions of each size class i is thus given by:

$$f_i = \mu_P \cdot F(T) \cdot H(I) \cdot U(L_i, N) - \mu_Z \cdot G(L_i, P_i) \cdot Z - V(L_i, M) - m_P - K,$$ (3)

where $\mu_P$ indicates the maximum growth rate and $F(T) = e^{0.063 \cdot T}$ is Eppley's formulation for temperature-dependent growth (Eppley, 1972). The light limiting term, $H(I)$, represents the total light I available in the upper mixed layer. According to Steele's formulation (Steele, 1962):

$$H(I) = \frac{1}{M(t)} \int_0^M \left[ \frac{I(z)}{I_s} \cdot e^{\left(1 - \frac{I(z)}{I_s}\right)} \right] dz,$$ (4)

where $I_s$ is the light level at which photosynthesis saturates and $I(z)$ is the irradiance at depth $z$. The exponential decay of light with depth is computed according to the Beer-Lambert law with a generic extinction coefficient $k_w$

$$I(z) = I_0 \cdot e^{-k_w \cdot z}.$$ (5)

The current version of our model does not specify any size-dependence for light absorption, although we provided suggestions on how this could be done (sections 4 and 6).

The nutrient limiting term U in equation 3 is determined by a Monod function with a half-saturation constant $K_N$, which scales allometrically with phytoplankton cell size L (Litchman et al., 2007),

$$U(L_i, N) = \frac{N}{N + K_N} = \frac{N}{N + (\beta_U \cdot e^{L_i \cdot \alpha_U})} , \qquad (6)$$

with $\beta_U$ and $\alpha_U$, respectively, intercept and slope of the $K_N$ allometric function (i.e., the power law $\beta_U \cdot S^{\alpha_U}$). This empirical relationship is based on observations of different phytoplankton groups (see figure 3b in Litchman et al., 2007), with the regression parameters rescaled from cell volume to ESD.

The loss term $G(L_i, P_i)$ in Equation 3, represents zooplankton grazing. As mentioned above, here we consider a single zooplankton population, which is assumed to be an assemblage of identical individuals with a size-selective feeding preference given by:

$$G(L_i, P_i) = \frac{e^{L_i \cdot \alpha_G}}{\sum_{i=1}^{n} P_i \cdot e^{L_i \cdot \alpha_G} + K_P} , \qquad (7)$$

where $\alpha_G$ is the slope for size-dependent grazing (or the power law $1 \cdot S^{\alpha_G}$) and $K_P$ is the half

saturation constant. This formulation is inspired by meta-analyses of laboratory data (Hansen et al., 1994, 1997), and reflects a grazing preference of zooplankton for smaller phytoplankton cells. For demonstration purposes, we use here a simple formulation for zooplankton grazing, however, other functional relationships can be implemented and tested in future versions of PhytoSFDM (see also sections 4 and 6).

The loss term $V(L_i, M)$ in Equation 3 represents the sinking of phytoplankton as a function of size and depth of the mixed layer,

$$V(L_i, M) = \frac{\beta_V \cdot e^{L_i \cdot \alpha_V}}{M(t)} , \qquad (8)$$

where the constants $\alpha_V$ and $\beta_V$ are the parameters of the function relating phytoplankton cell size to sinking velocity according to Stokes' law (Kiørboe, 1993), or the power law $\beta_V \cdot S^{\alpha_V}$. These

parameters are expressed here in units of meters per day.

Our model formulation does not specify an explicit size dependence for the phytoplankton maximum growth rate ($\mu_P$). Various compilations of data from laboratory experiments reveal different size scalings for $\mu_P$, either as a power law of cell volume (Litchman et al., 2007; Edwards et al., 2012) or as a uni-modal distribution in terms of cell size (Wirtz, 2011; Ward, 2012; Marañón et al.,

2013). Therefore, we adopted an approach similar to that of Smith et al. (2015), who reproduced

the uni-modal distribution of realised growth rate over size using two physiological trade-offs. We specified our trade-off in terms of three allometric relationships, and this results in an indirect size-dependence of phytoplankton growth rate.

The loss term $m_P$ in Equation 3 accounts for all phytoplankton losses other than those from grazing and mixing.

Differential equations for the nutrient (N), zooplankton (Z), and detritus (D) complete the model system:

$$\frac{dN}{dt} = -\sum_{i=1}^{n} \mu_P \cdot F(T) \cdot H(I) \cdot U(L_i, N) \cdot P_i \, + \, \delta_D \cdot D \, + \, K \cdot (N_0 - N) \,, \tag{9}$$

$$\frac{dP}{dt} = \sum_{i=1}^{n} (\mu_P \cdot F(T) \cdot H(I) \cdot U(L_i, N) \, - \, \mu_Z \cdot G(L_i, P_i) \cdot Z \, - \, V(L_i, M) \, - \, m_P \, - \, K) P_i \,, \tag{10}$$

$$\frac{dZ}{dt} = \sum_{i=1}^{n} \delta_Z \cdot \mu_Z \cdot Z \cdot G(L_i, P_i) \cdot P_i \, - \, m_Z \cdot Z^2 \, - \, K_Z \cdot Z \,, \tag{11}$$

$$\frac{dD}{dt} = \sum_{i=1}^{n} (1 - \delta_Z) \cdot \mu_Z \cdot Z \cdot G(L_i, P_i) \cdot P_i \, + \, \sum_{i=1}^{n} m_P \cdot P_i \, + \, m_Z \cdot Z^2 \, - \, \delta_D \cdot D \, - \, K \cdot D \,, \tag{12}$$

where $\delta_D$ is the mineralization rate and $N_0$ is the nitrogen concentration below the upper mixed layer. $\mu_Z$, $\delta_Z$ and $m_Z$ are, respectively, maximum growth rate, prey assimilation coefficient, and mortality rate of zooplankton. All parameter values and their units are reported in Table 1.

### 2.1.3 Dynamics of the aggregate phytoplankton community

The phytoplankton community comprising many distinct morphotypes (Equations 2 to 8) can be approximated with the so-called moment-based approach (Wirtz and Eckhardt, 1996; Norberg et al., 2001; Merico et al., 2009; Terseleer et al., 2014; Acevedo-Trejos et al., 2015). Wirtz and Eckhardt (1996); Norberg et al. (2001) and Merico et al. (2009), used a Taylor expansion together with a moment closure technique to approximate the whole community with three macroscopic properties, which correspond to the first three order moments of the approximated biomass distribution. These properties are: total biomass, mean trait, and trait variance. These works (Wirtz and Eckhardt, 1996; Norberg et al., 2001; Merico et al., 2009) were inspired by earlier applications in quantitative genetics (Abrams et al., 1993) and are reviewed by Smith et al. (2011) and more recently by Bonachela et al. (2015).

Here the whole phytoplankton community is characterised by the morphological trait cell size and by a trade-off that emerges from three allometric relationships described by Equations 6-8). The equations of the respective macroscopic properties are:

$$\frac{dP}{dt} \approx P \cdot (f + \frac{1}{2} \cdot f^{(2)} \cdot V), \tag{13}$$

$$\frac{d\overline{L}}{dt} \approx f^{(1)} \cdot V, \tag{14}$$

$$\frac{dV}{dt} \approx f^{(2)} \cdot V^2, \tag{15}$$

where f is the net growth rate (or the fitness function, see equation 3), and $f^{(n)}$ is the nth derivative of the net growth with respect to the trait. Due to competitive exclusion, however, the phytoplankton community loses functional diversity over time, i.e. the variance declines to zero with time, in both full and aggregate model formulations (Merico et al., 2014). We name this standard formulation "Unsustained Variance".

Alternatively, one can use the approximated model to focus only on changes in the mean trait, thus ignoring changes in the variance by fixing it to an arbitrary constant value:

$$\frac{dP}{dt} \approx P \cdot (f + \frac{1}{2} \cdot f^{(2)} \cdot V), \tag{16}$$

$$\frac{d\overline{L}}{dt} \approx f^{(1)} \cdot V, \tag{17}$$

$$\frac{dV}{dt} = 0. \tag{18}$$

While using these two formulations (i.e. Unsustained and Fixed Variance) can be acceptable in some special cases (e.g. in experiments that lead to competitive exclusion or where diversity is being manipulated), it is clear that they fail to account for changes in the adaptive capacity of the community, which requires allowing the size variance, and thereby functional diversity, to vary over time (Merico et al., 2014).

Within our modelling tool we also provide two alternative ways of treating the size variance: immigration (following Norberg et al., 2001) and trait diffusion (following Merico et al., 2014). The treatment with immigration considers the introduction of biomass and new trait values from hypothetical adjacent communities into the resident community. The addition of incoming amount of biomass per day is named immigration I,

$$\frac{dP}{dt} \approx P[f + \frac{1}{2} \cdot f^{(2)} \cdot V] + I, \tag{19}$$

$$\frac{d\overline{L}}{dt} \approx f^{(1)} \cdot V + \frac{I}{P}(L_I - \overline{L}), \tag{20}$$

$$\frac{dV}{dt} \approx f^{(2)} \cdot V^2 + \frac{I}{P}[(V_I - V) + (L_I - \overline{L})^2], \tag{21}$$

where $L_I$ and $V_I$ are, respectively, the mean size and the size variance of the immigrating community. As implemented by Acevedo-Trejos et al. (2015), we treat I as a density-dependent process (i.e.

I $= \delta_I \cdot P$), and set $L_I$ equal to the mean size of the resident community (i.e. $L_I = \overline{L}$). Thus, we assume that phytoplankton immigrating from adjacent areas are characterised by sizes similar to the simulated community, implying that the immigrating community has been exposed to the same selection pressures as the simulated community (Terseleer et al., 2014). We also assume that the rate of immigration increases proportional to the concentration of phytoplankton, consistent with observations of diversity patterns along the Atlantic Ocean (Chust et al., 2013).

The treatment of the size variance based on trait diffusion (Merico et al., 2014) gives:

$$\frac{dP}{dt} \approx P \cdot [f + \frac{1}{2} \cdot f^{(2)} \cdot V + \frac{1}{2} \cdot \nu(r_4 \cdot V - 3 \cdot r_2)], \tag{22}$$

$$\frac{d\overline{L}}{dt} \approx f^{(1)} \cdot V + \nu(r_3 \cdot V - 3 \cdot r_1), \tag{23}$$

$$\frac{dV}{dt} \approx f^{(2)} \cdot V^2 + \nu(r_4 \cdot V^2 - 5 \cdot V \cdot r_2 + 2 \cdot r), \tag{24}$$

where $\nu$ is the trait diffusivity parameter, $r$ is the reproduction rate (or gross growth), and $r_n$ is the nth derivative of gross growth with respect to the trait. Note that the process of trait diffusion (last term in the equation 24) depends on the gross growth $r$, via the trait diffusivity constant $\nu$, thus, an increase in phytoplankton gross growth causes an increase in trait variance (Merico et al., 2014).

The system of differential equations for all variance treatments is completed by equations describing gains and losses in nitrogen (N), zooplankton (Z), and detritus (D):

$$\frac{dN}{dt} = -\mu_P \cdot F(T) \cdot H(I) \cdot U(\overline{L}, N) \cdot P + \delta_D \cdot D + K \cdot (N_0 - N), \tag{25}$$

$$\frac{dZ}{dt} = \delta_Z \cdot \mu_Z \cdot Z \cdot G(\overline{L}, P) \cdot P - m_Z \cdot Z^2 - K_Z \cdot Z, \tag{26}$$

$$\frac{dD}{dt} = (1 - \delta_Z) \cdot \mu_Z \cdot Z \cdot G(\overline{L}, P) \cdot P + m_P \cdot P + m_Z \cdot Z^2 - \delta_D \cdot D - K \cdot D. \tag{27}$$

The first term in equation 25 represents a reduction of the nitrogen pool due to phytoplankton growth, which is a function of temperature, light, nitrogen, and mean size (see the description of equation 3 in the previous section). The last two terms in equation 25 represent sources of nitrogen due to remineralisation and mixing. The first term in equation 26 describes size-dependent grazing, while the last two terms describe losses of zooplankton due to mortality and mixing. The first term in equation 27 represents a fraction of phytoplankton biomass that is not assimilated by zooplankton and the following two terms represent the mortality of phytoplankton and zooplankton, respectively. The detritus pool is reduced by remineralisation and mixing. Parameter values and their units are reported in Table 1.

## 2.2 Environmental forcing

We compiled monthly climatological forcing data for mixed-layer depth (MLD), photosynthetic active radiation (PAR), sea surface temperature (SST) and concentration of nitrogen immediately below the upper mixed layer ($N_0$). The MLD data were obtained from Monterey and Levitus (1997) using the variable density criterion and are openly accessible from https://www.nodc.noaa.gov/OC5/ WOA94/mix.html. The PAR data were obtained from the Moderate Resolution Imaging Spectroradiometer (MODIS), for the time period 2002-2011. This dataset is managed and distributed by the NASA's Ocean Biology Processing Group (http://oceancolor.gsfc.nasa.gov/cms/). SST and $N_0$ were obtained from the World Ocean Atlas 2009 (WOA09), which is maintained and distributed by NOAA (https://www.nodc.noaa.gov/OC5/WOA09/pr_woa09.html). For consistency and efficiency, all data were transformed from their original formats (e.g. TXT and HDF) to NetCDF. All monthly forcing were spatially averaged over the selected location (square boxes in Figure 1) and then interpolated to obtain daily values (Figure 2).

## 3 Test-case simulation

A test-case model configuration is provided for a location of the North Atlantic ocean at 47.5° N 15.5° W (Figure 1), a region where seasonal changes in mean size and size diversity are well known (Acevedo-Trejos et al., 2015). This region presents the typical oceanographic conditions of a temperate environment (Figure 2). The environmental conditions produce a pronounced phytoplankton bloom in spring, which stimulates secondary production and almost the full depletion of nitrogen (Figure 3). Overturning of the water column in autumn restocks the pool of nitrogen and light limitation together with lower temperatures halts primary production (Figures 2 and 3).

### 3.1 Comparison of full and aggregate models

Within PhytoSFDM, we provide a practical example of how to implement and compare phytoplankton community models that aim to describe a) a full assemblage of species or morphotypes (see section 2.1.2), and b) an aggregate community (see section 2.1.3). The aggregate community model is an approximation of the full assemblage of species or morphotypes(Wirtz and Eckhardt, 1996; Norberg et al., 2001; Merico et al., 2009).

Figures 3 and 4 show the results of, respectively, the full model and the aggregate model for the Unsustained Variance case. N, P, Z, and D are unaffected by the type of model considered. As expected, the dynamics of P, $\overline{L}$, and V produced by the aggregate model are good approximations of those produced by the full model. Both models exhibit competitive exclusion, as indicated by the reduction in the number of morphotypes and consequently in the loss of size variance over time (Figure 4). The phytoplankton community evolves towards the optimal trait value, which is expressed by the fittest few morphotypes for the chosen parameterisation and the prevailing environmental

conditions. Although competitive exclusion is well established theoretically (Hardin, 1960), natural
communities of phytoplankton are typically very diverse, hence we will explore in the following the
effects of different ways of sustaining the variance.

## 3.2   Comparison of variance treatments

The key aspect of trait-based models is their ability to describe the phytoplankton community in
terms of mean trait and trait variance. Figures 5 and 6 show the results of one-year simulation after
an initial spin-up phase of four years. While the four treatments produce very similar, if not identical,
dynamics for N, P, Z, and D (Figure 5), the results for the mean size and the size variance differ
considerably among treatments (Figure 6).

   As already discussed, the system loses diversity over time when variance is unsustained. The loss
of diversity reduces the capacity of the community to adapt to changing environmental conditions
via shifts in species composition, as a flat year-round mean trait shows (Figure 6, grey lines). Under
Fixed Variance, size diversity is locked at an arbitrary value. If this value is high enough, the mean
size can adapt in response to changes in nutrient availability and grazing regimes (Figure 6). This
treatment can be useful for studies focusing only on the size structure of the community but it is
otherwise based on an arbitrarily fixed level of diversity and cannot offer meaningful insights, for
example about biodiversity and ecosystem functioning relationships.

   Trait Diffusion and Immigration show similar results for the mean size but not for the size variance
(Figure 6). Since the mechanism of Trait Diffusion depends on reproduction, i.e. gross growth (see
equation 24), the highest diversity of the community is reached in spring under high growth rates
and declines when moving towards winter. Size diversity also peaks in spring for the case of Immi-
gration because this mechanism is density-dependent (see equation 21), but the variances predicted
in autumn and winter are, respectively, lower and higher than those obtained with Trait Diffusion
(Figure 6). As mentioned above, this originates from the different assumptions underlying the Trait
Diffusion and Immigration treatments, which consider, respectively, an internal or an external source
of phytoplankton biomass, mean trait, and trait variance. In the case of trait-diffusion, such internal
source is gross growth because the size variance of the phytoplankton community is proportional
to it via the diffusivity constant $\nu$ (last terms in equations 22, 23 and 24). In contrast, immigration
represents a source of biomass (I) and size variance $(I/P[VI - V])$ external to the phytoplankton
community being simulated (e.g. from an adjacent patch). Hence, during the autumn-winter transi-
tion, the size variance tends to decline in the trait diffusion case as phytoplankton gross growth is
reduced by growth-limiting processes. Instead, the trait variance keeps building up to values similar
to the variance of the immigrating community in the case of Immigration.

### 3.3 Sensitivity to changes in parameter values

We tested the sensitivity of the annual mean in P, $\overline{\mathrm{L}}$, and V to variations of +/- 25 % in parameter values. To quantify this sensitivity, we formulated an index S that accounts for relative changes in model results:

$$S = \frac{X(p) - X(p')}{X(p)} \cdot 100 , \tag{28}$$

where $X(p)$ is the result of the state variable X obtained with the standard parameter p and $X(p')$ is the result of the state variable X obtained with the modified parameter $p'=p\pm25$ %.

The four treatments of size variance respond similarly to changes in parameter values (Figure 7). The annual means of all three state variables (P, $\overline{\mathrm{L}}$, and V) are sensitive to changes in the parameters controlling zooplankton grazing (i.e. $\mu_Z$, $m_Z$, $K_P$, $\delta_Z$). However, P also shows a sensitive response to parameters affecting phytoplankton gross growth, such as $k_w$, $I_s$, $\mu_P$, and $m_P$. Mean size is the most robust variable with less than 10 % relative change compared to the standard run. The size variance treatments for Immigration and Trait Diffusion are affected by the parameters controlling the input of exogenous (i.e. $\delta_I$ for Immigration) or endogenous variance (i.e. $\nu$, for Trait Diffusion). The results of the Unsustained Variance model are very sensitive to changes in $\mu_Z$ and the case of Fixed Variance shows a sensitivity that is similar to the other cases, except for the variance itself.

### 3.4 Computational efficiency

Trait-based models that aim at resolving the complexity of natural communities by incorporating many different species or functional types can be expensive in terms of computational time (Baird and Suthers, 2007; Follows et al., 2007; Bruggeman, 2007; Banas, 2011; Ward, 2012). Alternatively, trait-based models that focus on aggregate community properties such as total biomass, mean trait, and trait variance can be more computationally efficient. In table 2 we report a quantification of the computation time required for calculating the full and aggregate models presented here. We obtained more than 10-fold longer computation time for the full model than for the aggregate model. In addition, when we increase the resolution of the full model from 10 to 100 morphotypes, the difference in computation time increases by more than 20-fold. Thus increasing the realism, in terms of number of species or morphotypes comes at a significant computational cost.

## 4 Strength and weakness of moment-based approximations

Models are simplifications of reality and, as such, are based on assumptions. For example, the simple exponential growth model is based on a number of assumptions that do not hold in all circumstances (many factors affect the intrinsic growth rate, which is often not time-invariant, not all individuals within a population are identical, nothing can grow indefinitely, etc.). However, this model is widely

used within its range of validity. Likewise, the approximation of full models with moment-based approaches requires an assumption about the shape of the phytoplankton trait distribution (Wirtz and Eckhardt, 1996; Norberg et al., 2001; Merico et al., 2009). Typically, unimodal distributions, e.g. normal or log-normal, are assumed. However, depending on how the fitness function (i.e. the net growth rate of the phytoplankton community) is constructed and parameterised, the value of $f^{(2)}$, that is the rate of change of the variance (Equations 15, 18, 21 24), can be positive, implying a disruptive selection or branching. This represents an indication that the unimodality assumption does not hold (Bonachela et al., 2015). Alternatively, $f^{(2)}$ can remain negative over time, implying that the community continually loses variance, thus constituting a strong indication against the occurrence of disruptive selection. Therefore, models based on moment approximations require careful checks about the validity of the unimodality assumption throughout the time of the simulations. Figure 8 shows, for our test case, the predicted variance V, $f^{(2)}$ and its components for the four variance treatments. In our test case, $f^{(2)}$ is negative for all treatments and its changes are jointly driven by bottom-up, $f^{(2)} U(\overline{L}, N)$, and top-down processes, $f^{(2)} G(\overline{L}, P)$, i.e. the second derivatives with respect to the trait for nitrogen uptake (equation 6) and grazing (equation 7) terms. Sinking plays a role mainly during spring, but its influence is minor compared to the effects of nitrogen uptake and grazing.

It is unclear whether unimodality in size distributions is a robust feature in the oceans. Observational evidence from recent work (Downing et al., 2014) suggests that at large temporal scales, from 5 to 20 years, unimodality of size distributions is a consistent feature of phytoplankton communities of the North Sea. By contrast, multimodality is typically observed on temporal scales of less than one year (Downing et al., 2014). We consider that the observational evidence available remains insufficient to identify general patterns. However, the ocean is a highly variable environment and we considered it more likely that multimodality, for example because of size-selective grazing events, is a short-term, transient situation rather than the norm, because mixing would continuously reshuffle plankton assemblages and restore homogeneous conditions.

An aspect that our model does not include in its current version is the dependency of light acquisition on phytoplankton cell size. Given that the effect of cell size on light harvesting is well understood (Augusti, 1991; Finkel and Irwin, 2000; Finkel, 2001), it could be implemented in the model. Future versions of PhytoSFDM could address this gap by considering the vertical attenuation of light as a function of both phytoplankton biomass and cell size, following the approach proposed by Baird and Suthers (2007).

Uncertainty remains about how to described the zooplankton population, which we simplified as an assemblage of identical individuals. This has been the standard approach in plankton ecosystem modelling for decades and we based the first version of PhytoSFDM on this simple and classical formulation. In recent years, however, significant efforts have been made to increase the level of detail of the zooplankton component in ecosystem models. Approaches are numerous and include the

consideration of different zooplankton functional types, different size classes, and different feeding preferences and strategies (Banas, 2011; Ward, 2012; Prowe et al., 2012; Wirtz, 2012; Mariani et al., 2013; Vallina et al., 2014; Ryabov et al., 2015). A trait-based description of zooplankton can help in reducing model complexity while maintaining an adequate representation of diversity. The selection

of traits to consider for ecosystem models will depend on the questions under scrutiny. For example, traits that could characterize zooplankton-related process in ecosystem models that focus on nutrient cycling are maximum growth rates, stoichiometric requirements, and the size distribution of food particles (Litchman et al., 2013). Since many zooplankton traits scale allometrically with body size, scaling laws should be considered because they are effective ways to generalise the relation-

ships among different traits and thus to reduce model complexity (i.e. adds size-related functionality without the need for discretely parameterised zooplankton classes). Implementing such a diversity of grazing mechanisms and processes is a natural step forward in the development of ecosystem models. However, a consistent representation of different grazing strategies remains an aspect under development (Litchman et al., 2013; Smith et al., 2014). PhytoSFDM constitutes a starting model

platform for gradually building model complexity at different trophic levels.

## 5    Concluding remarks

Biological communities are complex adaptive systems (Levin, 1998) characterised by many components and interconnections that lead to emergent properties and nonlinear responses. Models help us to formalise and simplify the complexity we observe in nature. This simplification allows us to

render natural phenomena treatable and testable (Levins, 1966; Anderson, 2005, 2010). Over time, however, phytoplankton models have grown more complex, computationally more complicated, and often inaccessible to the wider scientific community, aspects that can all hamper advancements in the field. To help reverse this trend we developed PhytoSFDM as a tool to promote the use of trait-based models (whether species-explicit or aggregate models) of marine ecosystems.

A key decision in modelling is choosing an appropriate level of detail for the problem at hand. For example, a species-explicit model offers obvious advantages, which aggregate models cannot offer, when the interest lies in understanding the relative importance of particular species in providing certain ecological services or in quantifying the effect of disruptive selection. Aggregate models, instead, can be more useful at a higher level of abstraction, when the interest lies on macroecological

properties. In addition, as we have shown, aggregate models present an advantage with respect to computation time when compared to full models. The advantages in terms of reducing complexity and computation time remain unproven in spatially explicit settings (e.g. in 1D and 3D), although preliminary applications have shown promising results (Bruggeman, 2009).

     PhytoSFDM provides a set of methods, under the open source concept, to quantify macroecolog-

ical properties of phytoplankton communities, as an alternative to the traditional discrete, species-

explicit approach. This effort, we hope, will foster our understanding about the role that biodiversity plays in shaping marine ecosystems.

## 6  Code availability

PhytoSFDM is written in Python (version 2.7.x) as a lightweight and user-friendly package to facilitate use and re-distribution. We provide PhytoSFDM as free software under the GNU General Public License version 2. The python package is hosted in: a) GitHUB (https://github.com/SEGGroup/PhytoSFDM) a software repository that allows for version control, b) Zenodo (https://zenodo.org/record/49849) an open scientific repository, and c) PyPI (https://pypi.python.org/pypi/PhytoSFDM) one of the most popular python package repository. To be able to install and operate the package, the user should be familiar with the Python language and should have a running python distribution (preferably version 2.7.x) that includes the latest versions of the libraries *pip* and *setuptools*. Additional required libraries are *matplotlib*, *numpy*, *scipy* and *sympy*. PhytoSFDM can then be conveniently installed by typing the following command from a terminal window:

```
$ pip install PhytoSFDM
```

or by downloading the tarball from the GitHub repository. This is installed using the source file *setup.py* contained in the PhytoSFDM folder by typing:

```
$ python setup.py install
```

The package consists of three main modules: *Example*, *SizeModels*, and *EnvForcing*. *Example* is the entry point: it computes and compares full and aggregate models with the four treatments of variance (unsustained, fixed, trait diffusion, and immigration) at the testing location in the north Atlantic Ocean (centred at 47.5° N and 15.5° W). The example is run from a terminal by typing:

```
$ PhytoSFDM_example
```

or from an interactive python shell by typing:

```
>>> import phytosfdm.Example.example as exmp
>>> exmp.main()
```

The module *SizeModels*, contains the model variants. Here the user can a) modify the default parameters, b) symbolically solve the derivatives with respect to the trait, and c) log-transform mean trait and trait variance. To run the model at a specific location in an interactive python shell one should type:

```
>>> from phytosfdm.SizeModels.sizemodels import SM
>>> Lat=47.5
>>> Lon=344.5
>>> RBB=2.5
```

```
>>> SM1= SM( Lat , Lon ,RBB, " Imm " )
```

In the above example, the model is executed at a location in the North Atlantic Ocean centred at 47.5°N and 15.5°W (here transformed to a scale of 0° to 360°). RBB specify the range of the bounding box (in degrees) for averaging the environmental forcing variables. The fourth argument SM1 is an object that contains the call of the function SM, which runs the size model at the specified location and with the desired treatment for the size variance, in this case Immigration.

The last module, *EnvForcing*, consists of a class containing spatially averaged forcing data. The climatological data have monthly resolution but we include a method to interpolate the data to a daily time step. Spatially averaged and temporally interpolated forcing at a specific location can be extracted by typing:

```
>>> MLD= ExtractEnvFor ( Lat , Lon ,RBB, ' mld ' )
```

Additional information on the usage of the package is contained in the Readme file and in the repository webpage in GitHUB. The source code of our model is fully and freely accessible. Users can modify or add new model variants. This can be done by manipulating the module *sizemodels*, which contains model variants as separated methods within the class SM. By using a version control system such as GitHUB, users can fork our repository, i.e. create a copy, which allows one to freely change and experiment without affecting the original code. Users can also modify the original code and submit a new version by pulling a request. More details can be found in our GitHUB repository (https://github.com/SEGGroup/PhytoSFDM).

*Acknowledgements.* We would like to thank Jorn Bruggeman for his support on earlier versions of the model and for his suggestions while we were preparing the draft of this manuscript. EA-T and AM are supported by the German Research Foundation (DFG), through the priority programme DynaTrait (DFG-Schwerpunktprogramm 1704, subproject 19). SLS received support from the Japan Science and Technology Agency (JST) through a CREST Project. We are also grateful to Andrew Yool, Mark Baird, and an anonymous reviewer whose constructive suggestions helped improve our manuscript.

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

# Figures

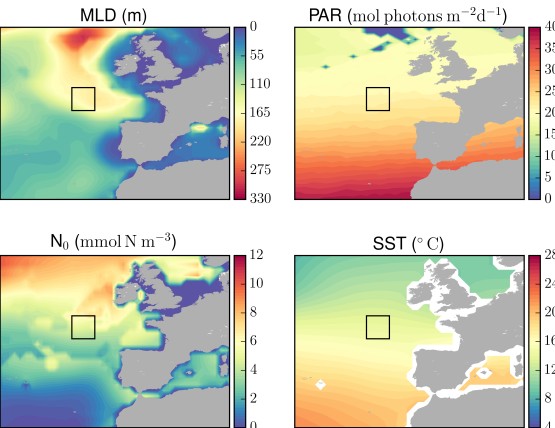

**Figure 1.** Environmental forcing variables considered in PhytoSFDM. The data shown are annual average of mixed-layer depth (MLD), photosynthetic active radiation (PAR), sea surface temperature (SST), and nitrogen concentration below the mixed-layer ($N_0$). The square boxes mark the location of the test-case simulation.

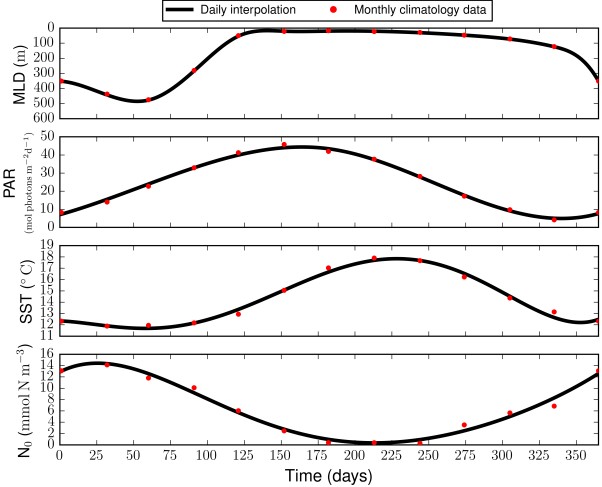

**Figure 2.** Temporal variation of the environmental variables. The monthly climatology data (red dots) are spatially averaged over the test location (square boxes in Figure 1). The interpolation (continuous line) is obtained with a $3^{rd}$ (MLD and PAR) and a $5^{th}$ (SST and $N_0$) order polynomial.

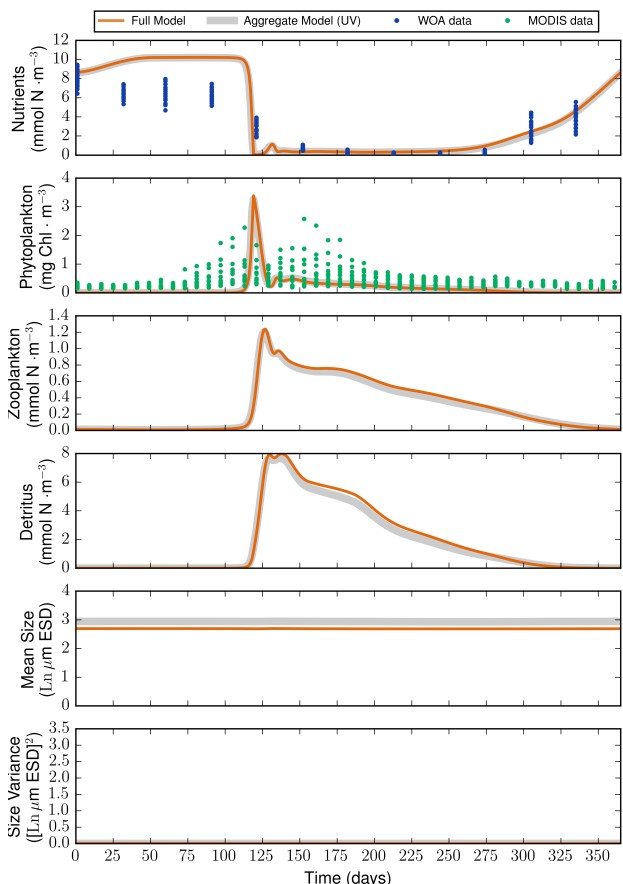

**Figure 3.** NPZD dynamics of the full model (Section 2.1.2 ) and of its equivalent aggregate model (Unsustained Variance, Section 2.1.3) for the last year of the simulations. The total phytoplankton in the full model corresponds to the sum of all $P_n$. The red dots are observations of nitrogen concentrations (monthly data obtained from the World Ocean Atlas) and the green dots are remotely sensed Chl-a data (8-day composite obtained from MODIS).

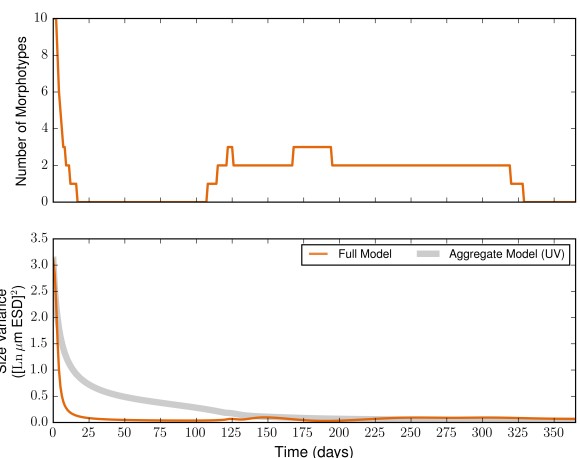

**Figure 4.** Number of morphotypes and size variance over the first year of the simulation. Here we included the morphotypes with a biomass greater than $0.01\ \mathrm{mmol\,N\,m^{-3}}$. Models that do not consider a mechanism to sustain variance exhibit competitive exclusion of morphotypes and a rapid decline of size diversity.

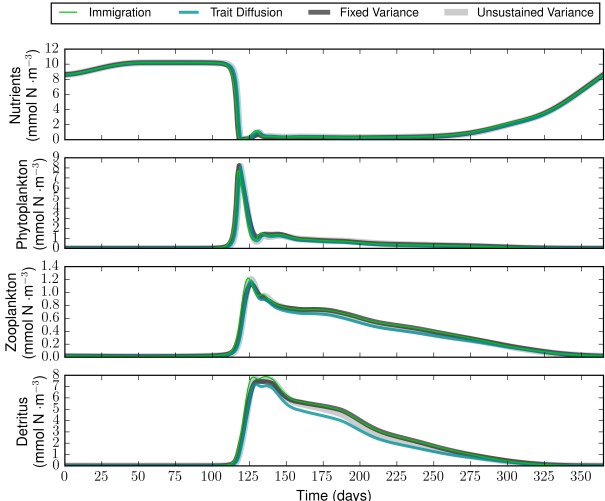

**Figure 5.** Nutrient, Phytoplankton, Zooplankton and Detritus dynamics over a seasonal cycle for the four variants of the aggregate model (see section 2.1.3), named Unsustained and Fixed Variance, Trait Diffusion, and Immigration.

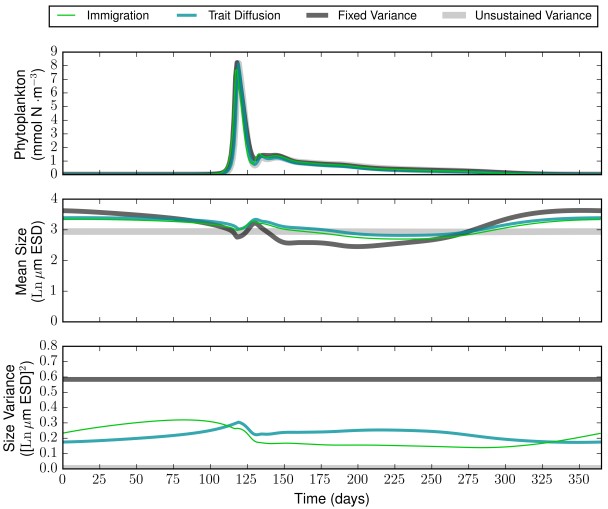

**Figure 6.** Dynamics of the size structured phytoplankton community and its functional size diversity for the four variance treatments (see section 2.1.3), named Unsustained and Fixed Variance, Trait Diffusion, and Immigration.

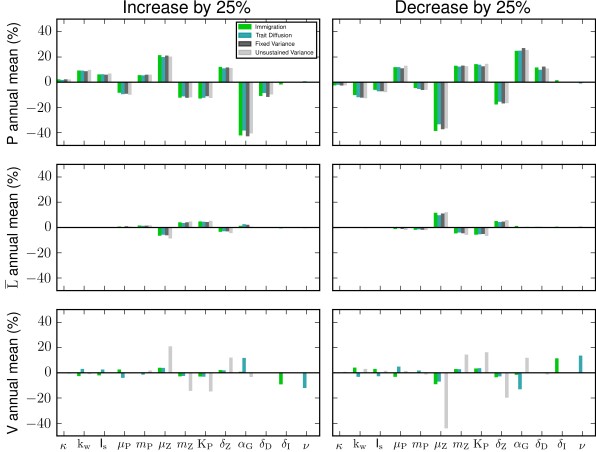

**Figure 7.** Sensitivity of four variance treatments to an increase and a decrease by 25 % in the default parameter values. The values and definitions of all parameters are given in table 1.

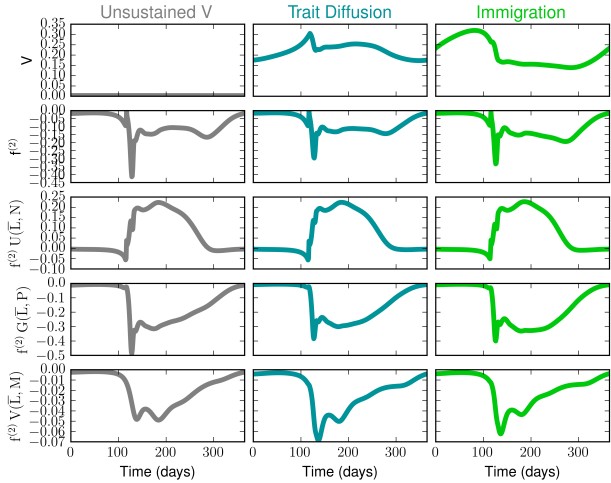

**Figure 8.** Components of the size variance (V), where $f^{(2)}$ is the second derivative of the fitness function with respect to the trait, $f^{(2)} U(\overline{L}, N)$, $f^{(2)} G(\overline{L}, P)$ and $f^{(2)} V(\overline{L}, M)$ are, respectively, nitrogen uptake, zooplankton grazing and phytoplankton sinking components of $f^{(2)}$.

**Tables**

**Table 1.** Parameters definitions, their units and their default values as provided in PhytoSFDM.

| Definition | Symbol (Units) | Value |
|---|---|---|
| Diffusive mixing across the thermocline | $\kappa$ (m·d$^{-1}$) | 0.1 |
| Light attenuation constant | $k_w$ (m$^{-1}$) | 0.1 |
| Optimum irradiance | $I_s$ (E m$^{-2}$d$^{-1}$) | 30 |
| P max growth rate | $\mu_P$ (d$^{-1}$) | 1.5 |
| P mortality rate | $m_P$ (d$^{-1}$) | 0.05 |
| Z grazing rate | $\mu_Z$ (d$^{-1}$) | 1.35 |
| Z mortality rate | $m_Z$ (d$^{-1}$) | 0.3 |
| P half-saturation | $K_P$ (mmol N m$^{-3}$) | 0.1 |
| P assimilation coefficient | $\delta_Z$ (-) | 0.31 |
| Mineralization rate | $\delta_D$ (d$^{-1}$) | 0.1 |
| Immigration rate | $\delta_I$ (mmol N d$^{-1}$) | 0.008 |
| Trait diffusivity parameter | $\nu$ (-) | 0.008 |
| Slope for allometric grazer preference | $\alpha_G$ ([$\mu$m ESD]$^{-1}$) | -0.75 |
| Intercept of the $K_N$ allometric function | $\beta_U$ (mmol N m$^{-3}$) | 0.14257 |
| Slope of the $K_N$ allometric function | $\alpha_U$ (mmol N m$^{-3}$ [$\mu$m ESD]$^{-1}$) | 0.81 |
| Intercept of the V allometric function | $\beta_V$ (m·d$^{-1}$) | 0.01989 |
| Slope of the V allometric function | $\alpha_V$ (m·d$^{-1}$ [$\mu$m ESD]$^{-1}$) | 1.17 |
| Size variance of immigrating P | $V_0$ (Ln [$\mu$m ESD]$^2$) | 0.58 |
| Number of morphotypes | n (-) | 10 or 100 |

**Table 2.** Computation time in seconds for the full model with 10 and 100 morphotypes and the four variants of the aggregate model.

| System | Full$_{10}$ | Full$_{100}$ | Unsustained | Fixed | Trait Diffusion | Immigration |
|---|---|---|---|---|---|---|
| MacOS 2.8 GHz Intel i7 | 282.515 | 6696.503 | 16.012 | 20.374 | 15.177 | 16.644 |
| Windows 3.0 GHz Intel i5 | 369.597 | 8722.753 | 20.236 | 22.485 | 20.736 | 20.298 |