# Peer review of "PhytoSFDM version 1.0.0: Phytoplankton Size and Functional Diversity Model"

_Geoscientific Model Development, 2016_

## Referee Comment (RC1) · Anonymous Referee #1 · 25 Aug 2016

**General Comments:**

The manuscript presents a modeling framework for simulating phytoplankton community structure. There is a brief description of a full 'discrete' model and of four simpler approximations of that model. This is followed by a comparison between the full model and the four approximations.

The manuscript is well written and easy to follow. I found the presented results to be informative, and I believe the manuscript merits publication in GMD, after some modifications.

My main concern is that the discrete model represents a somewhat easy target for the moment-based approximations. This is a problem, because the impression given by the authors is that the approximations are better in a number of ways relative to

the discrete model. This is not only because the approximations are computationally cheaper, but also because they can sustain more diversity. My concern is that they can only do this by adding arbitrary processes that were expressly designed to sustain diversity. The moment-based approximations are also incapable (at the moment) of representing some processes that can be relatively easily incorporated into the discrete model.

To give an example, the discrete model could (relatively) easily be extended to include a diverse zooplankton population, and this would increase diversity as an emergent property. It seems that the moment-based approximations would fare pretty poorly in this case, because the resultant (multimodal) biomass distribution would not be well approximated by the total biomass, mean cell size and variance of cell size.

I realize that accounting for this sort of disruptive selection is a major challenge with the moment-based approach, and I am not suggesting the authors need to account for it. I would, however, like them to acknowledge that it is a challenge. At the moment the Discussion section ("strengths and weaknesses of moment-based approximations") is very general, and a bit too philosophical. I would prefer if the authors focused on specific examples, and also tried to identify what challenges remain.

**Specific comments:**

Line 88 - "We define n . . . phytoplankton types". Which value of n did you use?

Line 90 – "The distribution of phytoplankton cell size" Would it be better to say "The distribution of biomass along the size dimension", or words to that effect? Please also specify here that you define only one nutrient, one zooplankton, and one detritus.

Line 93 – "constrains" should be "constraints".

Line 93 – "limits" should be "limit".

Line 94 – "more than 100 $\mu$m" How much more?

Eqn. 3 – Is the maximum growth rate size-independent? If so, this seems like a strange decision. Please explain.

Eqn. 6 – alpha and beta have different subscripts in the equation relative to the definition that follows.

Eqns. 6 – 8: I apologise for my ignorance, but does the form $\alpha e^{\log(size)\beta}$ correspond to a power law in your log-transformed formulation? If yes, please say so, if no, please say why you are using a different form.

Eqn. 7 – please specify that there is one zooplankton population that eats all phytoplankton. Please also add some justification/explanation for this choice.

Line 176 – "we treat I as a density-dependent process" Please explain why.

Eqns. 24-26. I think these equations need a little unpacking. It is a bit hard to see where the come from. For example, they look like they are based on the mean size phytoplankton, but I cannot see where (or how, and indeed if) the variance comes into it. Please try to make this a bit clearer.

Line 191 – "concentration of nutrients below the upper mixed layer ($N_0$)". Please define this more clearly. A first guess would suggest the average concentration between the ML and the seabed, but this doesn't make sense, because $N_0$ goes to zero in the summer.

Line 208 – Temperature would probably also be limiting in winter.

Lines 246 – 248: This stuff about exogenous and endogenous sources of biomass needs further explanation if I am to believe it is the fundamental cause of the observed differences.

---

## Referee Comment (RC2) · M. Baird (Referee) · 5 Sep 2016

The use of trait-based modelling has been one of the main approaches to tackling complexity in pelagic ecosystems, and size is undoubtedly the most important trait. This software will provide users with the opportunity to quickly explore size in modelling pelagic ecosystems. Through the provision of means to control the trait changes through either Immigration and/or adaptation, the tool also provides a step towards applying this approach in more complicated settings. Thus this manuscript meets the goals of GMD.

The most significance weakness of the PhytoSFDM model appears to be the lack of consideration of the effect of phytoplankton size on light capture by phytoplankton. This is especially surprising since the effect of size on light capture is arguably better understood than nutrient uptake and grazing due to the strongly geometric nature of the phenomena, and the Finkel paper they cite is a good starting place for this understanding. Should this angle be pursued, considering the vertical attenuation as a function of both phytoplankton biomass and the size-distribution could be easily undertaken (see Baird et al., 2007 for a simple to size-dependence, although more complicated version approaches to size are available).

But the impact of size on light capture is not the only trait that new users might explore. The authors invite readers to use their software. Many will want to add another process / complexity. It would be useful if Section 6 gave a few insights into how possible this is, and how a new user should go about it.

The manuscript is not overly long. I would recommend that the moment approach be given a brief summary, as this manuscript, if it produces interest, will likely be read by some scientists unfamiliar with the approach. By matching a brief summary with the specific equations and test case results in this paper may be quite useful.

Minor comments. L13 "World's oceans" or perhaps "global ocean" L22 Mentioning the Levin triad, which I had not heard of, in the 2nd line of the Introduction is a bit obscure. L78 replace "mixing" with "exchange" L106 H(I) isn't quite the average light, because light above the saturation level appears to be discounted. L122 I think the Hansen studies are based on lab work (whereas to oceanography types observations imply in the ocean). L135 I would include the phytoplankton equation in the list to (1) make it complete (i.e. have N. P. Z and D altogether), and (2) because in it earlier form it is written as 1/P dP/dt, so conservation of mass is not immediately obvious. L185 I wasn't sure if the outcome of the trait diffusion parameter was greater if the growth rate was faster. Is that the outcome of the equation above? If so, sounds entirely reasonable and should be stated more explicitly.

Reference: Baird, M. E., P. G. Timko, L. Wu (2007) The effect of packaging of chlorophyll within phytoplankton and light scattering in a coupled physical-biological ocean

model. Marine and Freshwater Res. 58, 966-981.

---

## Author Comment (AC1) · 30 Sep 2016

We appreciated the constructive and insightful comments received by the reviewers. Below we explain how we have addressed each of the points raised (quoted in italics) and attached, as a supplement, a revised version of the manuscript.

**Referee No.1**

*General Comments: The manuscript presents a modeling framework for simulating phytoplankton community structure. There is a brief description of a full "discrete" model and of four simpler approximations of that model. This is followed by a comparison between the full model and the four approximations.*

*The manuscript is well written and easy to follow. I found the presented results to be informative, and I believe the manuscript merits publication in GMD, after some modifications. My main concern is that the discrete model represents a somewhat easy target for the moment-based approximations. This is a problem, because the impression given by the authors is that the approximations are better in a number of ways relative to the discrete model. This is not only because the approximations are computationally cheaper, but also because they can sustain more diversity. My concern is that they can only do this by adding arbitrary processes that were expressly designed to sustain diversity. The moment-based approximations are also incapable (at the moment) of representing some processes that can be relatively easily incorporated into the discrete model. To give an example, the discrete model could (relatively) easily be extended to include a diverse zooplankton population, and this would increase diversity as an emergent property. It seems that the moment-based approximations would fare pretty poorly in this case, because the resultant (multimodal) biomass distribution would not be well approximated by the total biomass, mean cell size and variance of cell size. I realize that accounting for this sort of disruptive selection is a major challenge with the moment-based approach, and I am not suggesting the authors need to account for it. I would, however, like them to acknowledge that it is a challenge. At the moment the Discussion section ("strengths and weaknesses of moment-based approximations") is very general, and a bit too philosophical. I would prefer if the authors focused on specific examples, and also tried to identify what challenges remain.*

Good points, thank you. We do not claim that the processes we have selected are the only mechanisms that can sustain the diversity of plankton communities. We are aware that other processes can promote and sustain diversity and we agree that grazing can impact on the diversity of phytoplankton and model results more broadly. However, due to the physiological and behavioural complexity of the zooplankton community, there is still debate about how to best represent grazing in plankton ecosystem models. Previous works have explored different approaches, including the consideration of distinct zooplankton functional types, size classes, life stages, feeding preferences, and feeding strategies (Banas, 2011; Ward, 2012; Prowe et al., 2012; Wirtz, 2012; Mariani et al., 2013; Vallina et al., 2014; Ryabov et al., 2015). Although our model

does not aim to solve such issues, in light of the reviewer?s comment, we realize that these aspects deserved more space than we devoted to it in the original version of the manuscript. We have therefore expanded section 4 (Strength and weakness of moment-based approximations) by including a discussion of these aspects (lines 362 to 380 in the revised version of the manuscript).

*Specific comments:*
*Line 88 - "We define n : : : phytoplankton types". Which value of n did you use?*

We used n of 10 and 100. We have now specified this in the text (lines 91-92 in the revised manuscript) and added the description and values of the parameter n in Table 1.

*Line 90 - "The distribution of phytoplankton cell size" Would it be better to say "The distribution of biomass along the size dimension", or words to that effect? Please also specify here that you define only one nutrient, one zooplankton, and one detritus.*

Agreed, we have changed the relevant text as suggested and explained that we use only one nutrient, nitrogen, one zooplankton population which we assumed to be composed by identical individuals, and a single detritus pool (lines 89-91 and 94 in the revised manuscript).

*Line 93 - "constrains" should be "constraints".*

Done.

*Line 93 - "limits" should be "limit".*

Done.

*Line 94 - "more than 100_m" How much more?*

Marañón (2015) and Andersen et al. (2015) suggest an upper limit for phytoplankton volume of 108 $\mu$m3, which corresponds to 575 $\mu$m ESD. We have added this in the revised manuscript (lines 97).

*Eqn. 3 - Is the maximum growth rate size-independent? If so, this seems like a strange decision. Please explain.*

Various compilations of data from laboratory experiments reveal different size scalings for the maximum growth rate, either as a power law of cell volume (Litchman et al., 2007; Edwards et al., 2012) or as a uni-modal distribution in terms of cell size (Wirtz, 2011; Ward, 2012; Marañón et al., 2013). This makes it difficult to decide precisely what dependence to assume for the maximum growth rate. Smith et al. (2015) showed that a model based on two physiological trade-offs can reproduce the uni-modal distribution of realised growth rate over size, even without assuming any explicit size dependence for the maximum growth rate. Similarly, our model includes three allometric relationships, which are related respectively to phytoplankton nutrient uptake, zooplankton grazing, and phytoplankton sinking, and we assume that the maximum growth rate of phytoplankton is not size-dependent. Nevertheless, as in Smith et al. (2015), the realised phytoplankton growth rate depends indirectly on size,

via the prescribed allometric relationships. In light of this comment, we have added a short explanation in the revised manuscript (lines 141-148).

*Eqn. 6 - alpha and beta have different subscripts in the equation relative to the definition that follows.*

Thank you for pointing out these inconsistencies. We have corrected them (lines 120 and 121 in the revised manuscript).

*Eqns. 6 - 8: I apologise for my ignorance, but does the form _elog(size)_ correspond to a power law in your log-transformed formulation? If yes, please say so, if no, please say why you are using a different form.*

Yes, it is a power law. We have clarified this in the revised version of the manuscript (lines 121, 129, and 139).

*Eqn. 7 - please specify that there is one zooplankton population that eats all phytoplankton. Please also add some justification/explanation for this choice.*

Thank you for pointing this out. In the revised manuscript (lines 125 and 134), we have explained our motivation for choosing a single zooplankton population, which is assumed to be composed of identical individuals.

*Line 176 - "we treat I as a density-dependent process" Please explain why. Eqns. 24-26. I think these equations need a little unpacking. It is a bit hard to see where the come from. For example, they look like they are based on the mean size phytoplankton, but I cannot see where (or how, and indeed if) the*

*variance comes into it. Please try to make this a bit clearer.*

Agreed, we have expanded the explanations about the treatment of immigration as a density-dependent process (lines 203 and 208 in the revised manuscript) and equations 24-26 (lines 222 and 230 in the revised manuscript).

*Line 191 - "concentration of nutrients below the upper mixed layer (N0)". Please define this more clearly. A first guess would suggest the average concentration between the ML and the seabed, but this doesn't make sense, because N0 goes to zero in the summer.*

In the revised manuscript (lines 233-234) we now write "concentration of nutrients immediately below the upper mixed layer (N0)".

*Line 208 - Temperature would probably also be limiting in winter.*

Yes, correct! We have explained this (line 251 in the revised manuscript).

*Lines 246 - 248: This stuff about exogenous and endogenous sources of biomass needs further explanation if I am to believe it is the fundamental cause of the observed differences.*

In light of the reviewer's comment, we realised that our initial description of the results obtained with Trait-diffusion and Immigration treatments was probably too ambiguous. The difference in the results between the two treatments rises from the internal or external source of phytoplankton biomass, mean trait, and trait variance. In the case of trait-diffusion, such internal mechanism is represented by gross growth because

trait variance (and hence trait diffusion) is proportional to it. In contrast, immigration represents the external mechanism, which adds biomass and size variance from a (hypothetical) community adjacent to the one being simulated. Hence, during the autumn-winter transition the size variance in the case of trait diffusion tends to decline as growth-limiting processes reduce phytoplankton gross growth. Instead, the size variance keeps building up to values similar to the variance of the immigrating community in the case of immigration. We have revised the relevant text to make these processes more clear (lines 270 to 297 in the revised manuscript).

**Referee No.2. Dr. Mark Baird**

*The use of trait-based modelling has been one of the main approaches to tackling complexity in pelagic ecosystems, and size is undoubtedly the most important trait. This software will provide users with the opportunity to quickly explore size in modeling pelagic ecosystems. Through the provision of means to control the trait changes through either Immigration and/or adaptation, the tool also provides a step towards applying this approach in more complicated settings. Thus this manuscript meets the goals of GMD. The most significance weakness of the PhytoSFDM model appears to be the lack of consideration of the effect of phytoplankton size on light capture by phytoplankton. This is especially surprising since the effect of size on light capture is arguably better understood than nutrient uptake and grazing due to the strongly geometric nature of the phenomena, and the Finkel paper they cite is a good starting place for this understanding. Should this angle be pursued, considering the vertical attenuation as a function of both phytoplankton biomass and the size-distribution could be easily undertaken (see Baird et al., 2007 for a simple to size-dependence, although more complicated version approaches to size are available). But the impact of size on light capture is not the only trait that new users might explore. The authors invite readers to use their software. Many will want to add another process / complexity. It would be useful if Section 6 gave a few insights into how possible this is, and how a new user should go about it. The manuscript is not overly long. I would recommend that the moment approach be given a brief summary, as this manuscript, if it produces interest, will likely be read by some scientists unfamiliar with*

*the approach. By matching a brief summary with the specific equations and test case results in this paper may be quite useful.*

Thank you for bringing these aspects to our attention. The moment-based approximation has been explained in detail by Wirtz and Eckhardt (1996), Norberg et al. (2001) and Merico et al. (2009) and by the review works of Smith et al. (2011) and Bonachela et al. (2015). However, we share your view that it would be appropriate to include a brief summary of this technique in our manuscript and we have done so (lines 164-171in the revised manuscript).

We also consider your comment about the possibility of adding other size-dependent process in our model very useful. Following your suggestion, we have added a paragraph in section 4 (Strength and weakness of moment-based approximations, lines 356-361) in which we mention the first steps that could be undertaken for adding the size dependence of light absorption.

Also, given that the model is fully and freely accessible via GitHUB, Zenodo and PyPI, users can access and modify the original code. In GitHUB, users can find more information about how to fork (i.e. create a copy of) our code repository. Forking a repository allow users to freely experiment with changes without affecting the original code. Users can also modify the original code and submit a new version by "pulling" a request. We have explained this more explicitly in section 6 (Code availability, lines 446-452).

*Minor comments.*
*L13 "World's Oceans" or perhaps "global ocean"*

Done.

*L22 Mentioning the Levin triad, which I had not heard of, in the 2nd line of the Introduction is a bit obscure.*

The Levins' triad is part of a concept proposed by the ecologist Richard Levins in a classic article about the strategy of model building (Levins, 1966). Levins argued that it is impossible for a model to simultaneously maximize three important requirements, namely precision, realism, and generality and discusses three strategies of model building that sacrifice, in turn, one requirement in favour of the other two. We have revised the relevant sentence in the revised manuscript to make these concepts less obscure (lines 22-24).

*L78 replace "mixing" with "exchange"*

Done.

*L106 H(I) isn't quite the average light, because light above the saturation level appears to be discounted.*

Thank you for pointing this out. We have revised the sentence, which now reads: "H(I) represents the total light I available in the upper mixed layer".

*L122 I think the Hansen studies are based on lab work (whereas to oceanography types observations imply in the ocean).*

We agree. In the revised manuscript (lines 130-134), we have specified that the works of Hansen are based on meta-analyses of laboratory data.

*L135 I would include the phytoplankton equation in the list to (1) make it complete (i.e. have N. P. Z and D altogether), and (2) because in it earlier form it is written as 1/P dP/dt, so conservation of mass is not immediately obvious.*

Done.

*L185 I wasn't sure if the outcome of the trait diffusion parameter was greater if the growth rate was faster. Is that the outcome of the equation above? If so, sounds entirely reasonable and should be stated more explicitly.*

Yes, trait diffusion increases when phytoplankton gross growth increases. The trait diffusivity parameter ($\nu$) is, however, fixed as it represents the proportionality constant between trait diffusion and gross growth. We have explained this aspect more clearly (lines 214-216 in the revised manuscript).

**References**

Andersen, et al.: Characteristic sizes of life in the oceans, from bacteria to whales. Ann. Rev. Mar. Sci. 8, 1-25, 2015.

Banas, N. S.: Adding complex trophic interactions to a size-spectral plankton model: Emergent diversity patterns and limits on predictability. Ecol. Modell. 222, 2663-2675, 2011.

Bonachela, J. A., Klausmeier, C. A., Edwards, K. F., Litchman, E. and Levin, S. A.: The role

of phytoplankton diversity in the emergent oceanic stoichiometry. J. Plankton Res. 38, 1021-1035, 2015.

Edwards, K. F., Thomas, M. K., Klausmeier, C. A. and Litchman, E.: Allometric scaling and taxonomic variation in nutrient utilization traits and maximum growth rate of phytoplankton. Limnol. Oceanogr. 57, 554-566, 2012.

Levins, R.: The strategy of model building in population biology. Am. Sci. 54, 421-431, 1966.

Litchman, E., Klausmeier, C. A., Schofield, O. and Falkowski, P. G.: The role of functional traits and trade-offs in structuring phytoplankton communities: scaling from cellular to ecosystem level. Ecol. Lett. 10, 1170-1181, 2007.

Norberg, J., Swaney, D. P., Dushoff, J., Lin, J., Casagrandi, R., and Levin, S. A.: Phenotypic diversity and ecosystem functioning in changing environments: a theoretical framework. Proc. Natl. Acad. Sci. 98, 11376-81, 2001.

Marañón, E., et al.: Unimodal size scaling of phytoplankton growth and the size dependence of nutrient uptake and use. Ecol. Lett. 16, 371-379, 2013.

Marañón, E.: Cell Size as a Key Determinant of Phytoplankton Metabolism and Community Structure. Ann. Rev. Mar. Sci. 7, 1-24, 2015.

Mariani, P., Andersen, K. H., Visser, A. W., Barton, A. D. and Kiørboe, T.: Control of plankton seasonal succession by adaptive grazing. Limnol. Oceanogr. 58, 173-184, 2013.

Merico, A., Bruggeman, J. and Wirtz, K.: A trait-based approach for downscaling complexity in plankton ecosystem models. Ecol. Modell. 220, 3001-3010, 2009.

Prowe, A. E. F., Pahlow, M., Dutkiewicz, S., Follows, M. J. and Oschlies, A.: Top-down control of marine phytoplankton diversity in a global ecosystem model. Prog. Oceanogr. 101, 1-13, 2012.

Ryabov, A. B., Morozov, A. and Blasius, B.: Imperfect prey selectivity of predators promotes biodiversity and irregularity in food webs. Ecol. Lett. 18:1262-1269, 2015.

Smith, S. L., Pahlow, M., Merico, A. and Wirtz, K. W.: Optimality-based modeling of planktonic organisms. Limnol. Oceanogr. 56, 2080-2094, 2011.

Smith, S. L. et al.: Flexible phytoplankton functional type (FlexPFT) model: size-scaling of traits and optimal growth. J. Plankton Res.,38(4), 977-992, 2015.

Vallina, S. M., Ward, B. A., Dutkiewicz, S. and Follows, M. J.: Maximal feeding with active prey-switching: A kill-the-winner functional response and its effect on global diversity and biogeography. Prog. Oceanogr. 120, 93-109, 2014.

Ward, B. A., Dutkiewicz, S., Jahn, O. and Follows, M. J.: A size-structured food-web model for

the global ocean. Limnol. Oceanogr. 57, 1877-1891, 2012.

Wirtz, K. W. and Eckhardt, B.: Effective variables in ecosystem models with an application to phytoplankton succession. Ecol. Modell. 92, 33-53, 1996.

Wirtz, K.: Non-uniform scaling in phytoplankton growth rate due to intracellular light and $CO_2$ decline. J. Plankton Res. 33, 1325-1341, 2011.

Wirtz, K. Who is eating whom?: Morphology and feeding type determine the size relation between planktonic predators and their ideal prey. Mar. Ecol. Prog. Ser. 445, 1-12, 2012.

---

## Author Response (AR2)

Dear Dr. Yool,

Thank you for your comments and we are glad that our contribution is being considered for publication in *Geoscientific Model Development*. Below we explain how we have addressed your points (quoted in italics).

*Line 115: "versions" should be "version"*
Done.

*Line 204: correct to "... from adjacent areas are characterised ..."*
Done.

*Line 369-370: amend to "A trait-based description of zooplankton can help in reducing model complexity ..."*
Done.

*Line 447: "my" should be "by"*
Done.

*Figure 1: I believe einsteins are abbreviated as "E" not "Ein"; also, mol photons are often preferred to einsteins*
We have now adopted "mol photons" throughout the manuscript.

*Figure 1: both leftmost panels are rather murky (e.g. coastlines are far less distinct than in the rightmost panels) – has something gone wrong?*
Thank you for pointing this out. We have revised the figure using a lighter color for land mass, which produces a clear coastline in all panels.

*Also, in the new text around trait-based zooplankton mentioned above (lines 369-376), there could be a little more clarity around why the trait-based approach "reduces complexity". Perhaps a bracketed remark along the lines of "(i.e. adds size-related functionality without the need for discretely parameterised zooplankton classes)" could be added to the end of the sentence on line 376. Might that help?*
Yes, indeed the suggested parenthetical sentence improves the clarity of our original statement. Thanks for your suggestion! We have added it to the latest version of the manuscript.